# Identification and Expression Analysis of NAC Gene Family in Weeping Trait of *Lagerstroemia indica*

**DOI:** 10.3390/plants11162168

**Published:** 2022-08-21

**Authors:** Cuihua Gu, Linxue Shang, Guozhe Zhang, Qun Wang, Qingqing Ma, Sidan Hong, Yu Zhao, Liyuan Yang

**Affiliations:** 1College of Landscape and Architecture, Zhejiang Agriculture & Forestry University, Hangzhou 311300, China; 2Zhejiang Provincial Key Laboratory of Germplasm Innovation and Utilization for Garden Plants, Zhejiang Agriculture & Forestry University, Hangzhou 311300, China; 3Key Laboratory of National Forestry and Grassland Administration on Germplasm Innovation and Utilization for Southern Garden Plants, Zhejiang Agriculture & Forestry University, Hangzhou 311300, China

**Keywords:** *Lagerstroemia indica*, NAC gene family, qRT-PCR analysis, weeping trait

## Abstract

*Lagerstroemia indica* is a widely used ornamental plant in summer gardens because of its desirable plant shape. The weeping traits of plants are related to secondary cell wall thickness and hormone signaling. NAC (NAM-ATAF1/2-CUC2), as one of the plant-specific transcription factors, is a switch for the secondary cell wall and also involved in leaf senescence, phytohormone signaling, and other growth processes. We identified a total of 21 *LiNAC* genes from the transcriptome data, which we divided into 14 subgroups and 2 groups. The physicochemical characteristics of amino acids, subcellular localization, transmembrane structure, GO and KEGG enrichment, and expression patterns were also examined. The qRT-PCR analysis showed that the expressions of *LiNAC8* and *LiNAC13* in upright *L. indica* ‘Shaoguifei’ and weeping *L. indica* ‘Xiariwuniang’ were significantly higher from the beginning to the end of growth stage (S1–S3), and the expressions of ‘Shaoguifei’ were always higher than those of ‘Xiariwuniang’. However, *LiNAC2* showed a downward trend in S1–S3 and the relative expression level of ‘Shaoguifei’ was lower than that of ‘Xiariwuniang’. It is hypothesized that these *LiNAC* genes may be involved in the regulation of weeping traits in *L. indica*. The results of this study provide a basis for analyzing the functions of *LiNAC* genes and help to explore the molecular regulatory mechanisms related to the weeping traits in *L. indica*.

## 1. Introduction

Weeping is a result of drooping branch development. The buds first develop upward, then as the tree grows, the apex of the branches shifts to growing downward. The reasons for the formation of the drooping branches are relatively convoluted, nevertheless. The development of drooping branches may be influenced by genetic make-up, hormones, and secondary growth, but it may also be intimately linked to certain external conditions, such as gravity, light, and supporting forces [1]. The weep gene mutation that led to the gravity anomaly in *Prunus persica* is what causes the weeping characteristic [2]. The weeping characteristics of *Salix babylonica* were brought on by a lack of mechanical support, brought on by an excessive trunk elongation [3]. For the weeping of *P. mume*, the secondary cell walls of branches in the pendant extension zone are thinner than those of upright branches [4]. To modulate the weeping phenotype, IAA and GA_3_ biosynthetic-pathway-related genes were discovered in *P. mume* [5,6]. Recently, Zheng et al. found that genes related to cell division, cell development, and plant hormones played an important role in the tortuous-branch phenotype of *P. mume* [7]. Overexpression of a secondary-wall-associated cellulose synthase gene (*PtdCesA8*) was shown to result in a weeping phenotype in *Populus tremuloides* [8]. Ornamental plants often attract attention because of their specific traits. A certain plant architecture, known as weeping, has significant aesthetic appeal. Therefore, exploring the causes of plant drooping is potentially valuable for cultivating ornamental plants.

The ability to synthesize proteins via biological activities is controlled by transcription factors and some cis-acting elements, which also govern the spatiotemporal expression of downstream genes and eventually have an impact on the growth and development of organisms [9]. The genes NAM (no apical meristematic tissue) from *Petunia hybrida* [10], *ATAF1/2* (*Arabidopsis* transcription activation factor) found in *Arabidopsis thaliana* [11], and *CUC2* (cup2 shaped cotyledon) from *A. thaliana* are the sources of the name *NAC* [11]. At the N-terminal end of the protein, 150 amino acids make up its conserved structural domain, including five substructural components (A, B, C, D, and E) [12]. According to multiple studies, *NAC* has a wide spectrum of regulatory effects on plant growth and development [13,14,15,16,17]. Therefore, it is potentially possible to probe the weeping properties of plants from *NAC*.

Many studies show that the *NAC* gene family affects the weeping of plants by controlling the synthesis of secondary cell walls [15,18]. Inhibiting the expression of *SND1* and *NST1* in *Arabidopsis* at the same time can limit the synthetic gene expression in the three components of secondary cell walls (cellulose, xylem, and lignin), causing the flower stem cells to droop and fail to create secondary walls [15]. Studies on *SND4* and *SND5* revealed similar functions [19]. While the secondary cell wall of the stem of the overexpressed *PtrWND1B-s* plants greatly thickened and allowed them to grow upright, the secondary cell wall of the overexpressed *PtrWND1B-l* plants was unable to do so and displayed a phenotype similar to that of *PtrWND1B-RNAi* [18]. *NAC15* was significantly expressed in the poplar xylem and is most likely the primary factor for transgenic tobacco to develop tall stems [20]. Lignin is an important component in secondary cell walls and *EjNAC1* is associated with fruit lignification by activating genes involved in lignin biosynthesis [21]. The main regulatory mechanism in melatonin in enhancing flower stem strength was also to promote lignin accumulation and changed S/G lignin ratio [22]. Co-expression analysis of *Pm024213* showed that most of the related genes were involved in auxin and lignin biosynthesis [23].

Because of its lovely blossoms and exquisite branches, the crape myrtle (*Lagerstroemia indica* L.), a deciduous shrub and small tree, has considerable ornamental value [24]. *L. indica* branches were classified into flat, upright, and weeping branches by Tian [25]. The study of the molecular mechanisms of *L. indica’s* weeping trait will benefit from an understanding of the *NAC* gene family function. However, nothing is known about the *NAC* gene studies in *L. indica*. In this study, the *NAC* genes in *L. indica* were discovered using a variety of bioinformatics techniques. Additionally, the physiological and biochemical properties of the LiNAC proteins were examined and a phylogenetic tree of the LiNAC proteins in *L. indica* and *A. thaliana* was created. To clarify whether *LiNAC* genes are involved in regulating the weeping trait of *L. indica*, we analyzed the relative expression levels of branches at different growth stages. Overall, the findings might support a theoretical hypothesis of *NAC* gene family in *L. indica*.

## 2. Results

### 2.1. Identification of NAC Genes in L. indica

Using AtNAC proteins as queries, 60 *LiNAC* genes were discovered by BLASTP searches in the *L. indica* transcriptome database and 43 putative *LiNAC* genes were discovered using an HMMER3 search in the *L. indica* protein database using the NAC-type NAM model (PF02365). Finally, using SMART and the NCBI Conserved Domain to remove duplicated genes and proteins without NAM-conserved domains, we were able to obtain 21 members of the *LiNAC* family. For further searching, all genes were given the designations *LiNAC1-LiNAC21* in the order of their assembled IDs (Table 1).

Proteins encoded by the 21 *LiNAC* genes contained 234 (LiNAC14) to 668 (LiNAC7) amino acids and their molecular weights (MWs) ranged from 26,491.05 (LiNAC14) to 74,336.43 (LiNAC7) Da, with an average molecular weight of 39,677 Da. Their predicted isoelectric point (pI) ranged from 4.42 (LiNAC4) to 9.93 (LiNAC19), with an average value of 6.90. Instability index calculation predicted that 17 (81%) of the NAC proteins were unstable. The LiNAC3, LiNAC4, LiNAC14, and LiNAC15 predicted proteins had instability indices less than 40 and were classified as stable proteins. Aliphatic amino acid indices showed that the thermal stability of the proteins ranged from 55.89 (LiNAC9) to 75.47 (LiNAC14). All LiNACs predicted protein grand average of hydropathicity (GRAVY) values as negative, demonstrating that they were predominantly hydrophilic. The predicted transmembrane structure results showed that LiNAC7 and LiNAC16 proteins had one transmembrane structure at the C-terminal of the 21 LiNAC proteins (Appendix A). Cell-PLoc subcellular localization predictions suggested that almost all LiNAC proteins were located in the Nucleus (Table 1).

### 2.2. Phylogenetic Analysis and Classification of LiNAC Proteins

We used the neighbor-joining (NJ) method to create a phylogenetic tree with 1000 bootstrap replications to shed light on the evolutionary relationships between proteins from *L. indica* and *A. thaliana*. Based on well-established *Arabidopsis* family classification and their homology with NAC proteins in *Arabidopsis*, 21 LiNAC proteins were classified into two groups, Group Ⅰ and Group Ⅱ, including 14 subgroups (Figure 1): OsNAC7, ANAC011, NAM, NAC1, ATAF, ANAC3/NAP, SENU5, ONAC022, TIP, NAC2, OsNAC8, ANAC001, ONAC003, and ANAC063. In our analysis, no LiNAC members from the subgroups ANAC011, NAM, ATAF, AtNAC3/NAP, OsNAC8, ONAC003, and ANAC063 were identified. Subgroup TIP had the most members (6) of *L. indica* and subgroups NAC1 and OsNAC7 had the fewest (1).

The LiNAC proteins clustered in the same subgroup, according to phylogenetic relationships, and may have related roles. For instance, NAC1 members are crucial for auxin signaling and the growth of lateral roots in plants [26]. The *NTL9* gene belonging to the TIP subgroup is heavily implicated in plant immune response [27]. The NAC family members in the ATAF, NAP, and AtNAC3 subgroups have a conserved role in stress response [28] and leaf senescence [17,29]. There are two distinct subgroups of NAC proteins that are involved in the formation of vascular vessels and the creation of secondary plant cell walls [30]. *SND2*, *SND3*, and other participants were placed into subgroup ONAC003, while *NST1*, *NST2*, *NST3/SND1*, and *VNDs* were grouped into subgroup OsNAC7.

### 2.3. Conserved Sequence Alignment and Motif Analysis

Multiple sequence alignments were created by Jalview for the 21 LiNAC proteins to investigate the presence and locations of conserved protein domains (Figure 2) and Weblogo displayed the conserved domains with five subdomains (A–E) (Figure 3). All LiNAC family members contained a NAM domain (Figure 4C), which contains a highly conserved A–E subdomain. To further analyze the structural diversity in LiNAC proteins, conserved motifs were searched using the MEME program. In total, 20 distinct motifs were identified (Figure 4B). Most LiNAC proteins contained motif 3 (representing subdomain A), motif 4 (representing subdomain B), motif 2 (representing subdomain C), motif 1 (representing subdomain D), and motif 5 (representing subdomain E) (Appendix A). Additionally, the majority of the members in the phylogenetic tree that were closely linked displayed comparable motifs in the same alignment and location (Figure 4A). Members in the TERN subgroup had nearly identical motifs. Interestingly, some motifs were identified in a certain subgroup. For example, compared with Group Ⅰ, the number of motifs in Group Ⅱ was small. Motif 14 was found only in the NAC1 subgroup and motif 16 in the ONAC022 subgroup. The results revealed that LiNAC proteins clustering in the same subgroup may have comparable biological roles and that specific motifs may be associated with specific functions of different subgroups.

### 2.4. Annotation and Enrichment Analysis of LiNAC Genes

GO functional annotation of 21 *LiNAC* genes revealed that 18 of them are involved in biological processes (BPs), cellular components (CCs), and molecular functions (MFs). We examined the GO enrichment data of 18 *LiNAC* genes to forecast their biological roles. The immunological effector mechanism has the highest degree of enrichment, as shown in Figure 5A, scoring 54.4, followed by the response of cells to cold stress, with an enrichment score of 51.72. Additionally, we found that many genes have roles in a variety of developmental processes (such as leaf development and shoot development).

As shown in Figure 5B, five *LiNAC* genes were enriched in the Nucleocytoplasmic Transport (ko03013), Phenylpropanoid Biosynthesis Pathway (ko00940), and Signal Pathway (ko04020), according to a KEGG pathway analysis of 21 *LiNAC* genes.

### 2.5. Determination and Analysis of Lignin Content

The lignin content in *L. indica* was measured by collecting the branches of ‘Shaoguifei’ and ‘Xiariwuniang’ at the early growth stage, the growth stage, and the end of the growth stage. The lignin content in the branches was analyzed separately for different stages and plants. Figure 6C shows that the lignin content in the branches of both ‘Shaoguifei’ and ‘Xiariwuniang’ increased gradually with the maturity of the growth period and the lignin content in ‘Shaoguifei’ was always higher than that in ‘Xiariwuniang’. We found that the difference in lignin content between the two branches was not obvious at the early growth stage, but there was a significant difference between the lignin content in ‘Shaoguifei’ and ‘Xiariwuniang’ at the growth stage and the end of the growth stage.

### 2.6. Expression Analysis of LiNAC Genes in Regulating the Weeping Trait

To identify which *LiNAC* genes were involved in regulating the weeping trait of *L. indica*, we used real-time PCR to analyze the expression patterns of *LiNAC* genes in branches of ‘Shaoguifei’ and ‘Xiariwuniang’ during the early growth stage, the growth stage, and the end of the growth stage. The expression pattern analysis is shown in Figure 7. The relative expression levels of *LiNAC3* and *LiNAC20* showed the same trend in the early growth stage (S1), growth stage (S2), and the end of the growth stage (S3) for ‘Shaoguifei’ and ‘Xiariwuniang’, which showed an increase first and then a decrease, and the expression level was higher in S2. The relative expression levels of five genes, *LiNAC7*, *LiNAC8*, *LiNAC14*, *LiNAC19*, and *LiNAC11*, in ‘Shaoguifei’ gradually increased from the S1 stage and reached the highest in S3. We found that, except for *LiNAC8* and *LiNAC11*, the relative expression of *LiNAC7*, *LiNAC14*, and *LiNAC19* showed a trend of increasing and then decreasing in ‘Xiariwuniang’, with the highest expression in the S2. Interestingly, the expression trend of *LiNAC8* in ‘Xiariwuniang’ is consistent with that in ‘Shaoguifei’, while the relative expression level of ‘Shaoguifei’ is slightly higher than that of ‘Xiariwuniang’ in three stages and the relative expression level is about twice that of ‘Xiariwuniang’. We noticed that the relative expression of nine genes (*LiNAC1*, *LiNAC9*, *LiNAC10*, *LiNAC16*, *LiNAC4*, *LiNAC5*, *LiNAC9*, *LiNAC21*, and *LiNAC15*) decreased significantly from S1 to S2 in ‘Shaoguifei’ and increased gradually in S3. Except for four genes, *LiNAC1*, *LiNAC10*, *LiNAC16*, and *LiNAC15*, whose relative expression decreased from S1 to S3 in ‘Xiariwuniang’, all the other five genes showed an increasing and then decreasing trend and their relative expression peaked at S2. The relative expression levels of *LiNAC2*, *LiNAC6*, *LiNAC12*, *LiNAC18*, and *LiNAC13* in ‘Shaoguifei’ showed a trend of gradual decline and the lowest expression level in S3 (except *LiNAC13*), especially *LiNAC12*, in S3 period, the expression level of *LiNAC12* was 0.1-times that of ‘Shaoguifei’. Interestingly, *LiNAC2*, *LiNAC18*, and *LiNAC13* also showed the same expression trend as that of ‘Shaoguifei’ in ‘Xiariwuniang’. Notably, the relative expression of *LiNAC2* in ‘Xiariwuniang’ was always higher than that of ‘Shaoguifei’, especially in the S1–S2 stages. Further, the expression of *LiNAC13* was always higher than that of ‘Xiariwuniang’ in all stages; the relative expression was 9.7-times higher than that of ‘Xiariwuniang’ in S3.

## 3. Discussion

### 3.1. The Characteristics of NAC Gene Family in L. indica

In this study, we identified 21 complete *LiNAC* genes with Open Reading Frames based on *L. indica* transcriptome data and analyzed their basic information. We predicted and analyzed 21 LiNAC proteins, such as isoelectric power, relative molecular weight, and transmembrane structure, LiNAC7, and LiNAC16 have a transmembrane structure at the C-terminal, and it is speculated that they are likely to be NAC-membrane-binding transcription factors. Almost all *LiNAC* genes are predicted to be localized in the nucleus (Table 1); *LiNACs* are probably mostly functional in the nucleus. The conserved structure and motif of *LiNACs* were also examined (Figure 4B) and it was shown that all of them possessed the normal NAM structure (Figure 4C). Nearly all *LiNAC* genes have motif 3, motif 4, and motif 1, although motif 2 is missing from Group II, which may be related to the loss of assembled data. The motifs found in the gene members of various evolutionary branches vary and motif 3, motif 4, motif 2, motif 1, and motif 5 may be the conserved motifs in the NAC family (Appendix A). Phylogenetic trees of *L. indica* and *Arabidopsis* divide into 2 groups with 14 subgroups (Figure 1) and *LiNAC* genes were not found in many subgroups (e.g., ANAC011, ATAF, and AtNAC3/NAP). Overexpression of *AtNAP* causes early senescence in *A. thaliana* and members of the ATAF subgroup have a significant role in leaf senescence and other characteristics of plants, indicating that these subgroups may be involved in the process of plant senescence. *LiNAC8* is closely connected to AT1G32770.1, AT3G61910.1, and AT2G46770, for instance, leading one to hypothesize about the function of the *LiNAC* gene based on genetic similarities. AT1G32770.1 (*SND1*) is a key gene that regulates the synthesis of cell secondary walls. In *Arabidopsis*, *SND1* is specifically expressed in xylem fibers, and functional silencing led to secondary wall thinning and the inability of pedicel fiber cells to form secondary walls and drooping. The other subgroups of the *LiNAC* gene may play different biological functions.

Further, 18 of the 21 *LiNAC* genes were shown to be engaged in the two biological processes of GO, leaf growth and branch creation, according to KEGG and GO enrichment analyses (Figure 5). While lignin, a phenylpropane-derived polymer, together with cellulose and hemicellulose, forms the cell wall of plant vascular tissue and provides mechanical support for plant upright growth, KEGG enrichment results revealed that *LiNAC* genes were involved in the phenylpropanoid biosynthesis pathway. As a result, it is hypothesized that the *LiNAC* genes may influence the production of secondary cell walls in plants to regulate the weeping trait.

### 3.2. LiNACs Are Closely Associated with the Weeping Trait of L. indica

It has been suggested that the weeping trait may be associated with abnormal GA signaling [31]. The drooping phenomenon also occurs when the gravity response changes [32]. Changes in the distribution of plant hormones often affect the formation of plant xylem and bast, which is another factor contributing to the phenomenon of weeping plants. Lignin is a phenylpropanoid-derived polymer that, together with cellulose and hemicellulose, forms the cell wall of plant vascular tissue and provides mechanical support for upright plant growth [33,34]. The NAC family of transcription factors is the master switch in the regulation of secondary cell wall thickening in plants and influences the plant phenotype by regulating the growth of secondary cell walls in plants. The cell wall thickness of upright branches in *P. mume* was significantly higher than that of weeping branches. In terms of overall expression, *PmWND1*(*PmNAC082*) [35] regulated downstream secondary-wall-synthesis-related genes, which were all significantly less expressed in the weeping branches than in the upright branches, indicating that *NAC* may play an important role in regulating the production of weeping traits in plants.

In this study, qRT-PCR analysis of upright *L. indica* ‘Shaoguifei’ and weeping *L. indica* ‘Xiariwuniang’ at three different growth stages showed that *LiNACs* played an important role in the weeping trait of *L. indica* and different *LiNACs* expression patterns were different during this process. The expression of *LiNAC13*, a member of the NAC1 subgroup, did not change significantly in the three growth stages of ‘Shaoguifei’, but decreased gradually in ‘Xiariwuniang’ and was much lower than that of ‘Shaoguifei’. The expression of *LiNAC13* in ‘Xiariwuniang’ decreased gradually and was much lower than that of ‘Shaoguifei’. NAC1 is normally involved in hormone signaling in plants to regulate plant growth and development [26] and it is speculated that *LiNAC13* may be involved in regulating weeping traits in plants by participating in the phytohormone signaling pathway, but this remains to be verified. The relative expression of *LiNAC2* in S1–S3 is always higher in ‘Xiariwuniang’ than in ‘Shaoguifei’, especially in the S1–S2 stages, and there is not much difference between them in the S3 stage. *LiNAC2* is closely related to AT3G105001/*NTL4*, so we hypothesized that *LiNAC2* began to negatively regulate the weeping trait of *L.*
*indica* at S1. This still needs to be further verified by subsequent experiments. *LiNAC12* is closely related to AT5G13180.1 (*VNI2*), a VNI2 transcription factor that plays a molecular linkage role between plant response to environmental stress and regulation of leaf longevity [36]. In contrast, the expression of *LiNAC12* in ‘Xiariwuniang’ was always higher than that of ‘Shaoguifei’ and reached a peak during S2. It is speculated that ‘Xiariwuniang’ may have an advantage over ‘Shaoguifei’ in regulating plant growth and coping with environmental stress during development, which may also be a reason for its weeping branch formation. The relative expression of *LiNAC8* (which is closely related to AT1G32770.1/*SND1*) gradually increased in the three growth stages (S1–S3) of ‘Shaoguifei’ and ‘Xiariwuniang’ branches and the expression of ‘Shaoguifei’ was always higher than that of ‘Xiariwuniang’. In our study, by measuring the lignin content in ‘Shaoguifei’ and ‘Xiariwuniang’ branches at different growth stages, we found that the lignin content in ‘Shaoguifei’ was always higher than that of ‘Xiariwuniang’. Further, in the early stage of growth, there was not much difference in the content between the two, but with the development, especially at the end of the growth stage, the lignin content in ‘Shaoguifei’ was much higher than that of ‘Xiariwuniang’ (Figure 6). Inhibition of *SND1* expression in *Arabidopsis* leads to a dramatic decrease in the expression of all three secondary wall synthesis genes and pedicel fiber cells are unable to form secondary walls and droop grow [15]. In addition, dominant inhibition of *PtrWNDs* in poplar caused significant thinning of secondary walls of transgenic poplar stem fibers and failure of plants to grow upright [18]. Lignin is an important component in the secondary cell wall of plants and when the lignin content increases, the fruit of *Eriobotrya japonica* also becomes hard [21]. Therefore, we speculate that ‘Shaoguifei’ has more lignin content than ‘Xiariwuniang’ and the branches of ‘Shaoguifei’ provide more mechanical support, while ‘Xiariwuniang’ has less lignin content and, therefore, droops. *SND5* in *Arabidopsis* is closely connected to *SND2/3*, which is involved with the secondary wall and its direct homolog in poplar, according to Zhong et al. [19]. Similar findings were found in our study, *LiNAC8* showed a consistent trend with lignin content at S1–S3 in different *L. indica* species, which indicated that *LiNAC8* may be involved in affecting the synthesis of secondary walls and determining the weeping trait of *L. indica*. However, this conclusion has to be further investigated.

## 4. Materials and Methods

### 4.1. Identification and Sequence Analysis of NAC Genes from L. indica

As a reference sequence, the NAC protein sequence was acquired from the *Arabidopsis* TAIR database (https://www.arabidopsis.org/) (accessed on 17 March 2022) and members of the putative *LiNAC* genes were sought for using the *L. indica* transcriptome database (the raw sequence data were obtained from the website https://bigd.big.ac.cn/gsa) (accessed on 17 March 2022) and a BLASTP search. Then, we used the local HMMER3.0 software(Robert, D.F.; Ashburn, VA, USA) and the Hidden Markov (HMM) profile of the NAC protein (PF02365), which was downloaded from the Pfam database (http://pfam.xfam.org) (accessed on 17 March 2022). The potential gene members discovered using the two search techniques were pooled. The conserved NAM domain was discovered in all potential *LiNAC* genes using the Batch Web CD-Search Tool (https://www.ncbi.nlm.nih.gov/cdd) (accessed on 17 March 2022) and SMART (http://smart.embl.de/) (accessed on 17 March 2022), ensuring the correctness of the results.

### 4.2. Characterization of LiNAC Proteins

The website ExPasy (https://www.expasy.org/) (accessed on 18 March 2022) predicted and examined the physicochemical characteristics of all LiNAC potential proteins, including molecular weight and theoretical points. TMHMM-2.0, a website that can be accessed online (https://services.healthtech.dtu.dk/service.php?TMHMM-2.0) (accessed on 18 March 2022), examined the transmembrane structure of LiNAC protein. The online program MEME (https://meme-suite.org/meme/tools/meme) (accessed on 20 March 2022) was used to find conserved motifs. The following parameters were utilized: maximum number of motifs = 20; search model = zero or one occurrence per sequence; default values for the remainder. TBtools software v1.098661(Chen, C.J.; Guangzhou, China) was used to display the results [37]. Cell-PLoc2.0 was used to create predictions for subcellular localization (http://www.csbio.sjtu.edu.cn/bioinf/plant-multi/#) (accessed on 20 March 2022).

### 4.3. Sequence Alignment and Phylogenetic Tree Construction

Using MUSCLE, the sequences of the found 21 LiNAC proteins and 80 AtNAC protein sequences were aligned. In MEGA 6.0, the outcome was utilized to create a neighbor-joining (NJ) phylogenetic tree using 1000 bootstrap repetitions. P-distance and pairwise were the predetermined parameters. Phylogenetic tree were illustrations created with iTOL (https://itol.embl.de/) (accessed on 20 March 2022). Using the Jalview software 2.11.2.4 (Andrew, M.W.; Cambridge, MA, USA) (http://www.jalview.org/)(accessed on 20 March 2022), the amino acid sequences of conserved domains were compared and modified and conserved motif Logos were produced using the WebLogo program (http://weblogo.threeplusone.com) (accessed on 20 March 2022).

### 4.4. Annotation and Enrichment Analysis in GO and KEGG Databases

Online software EggNOG-Mapper (http://eggnog-mapper.embl.de/) (accessed on 20 March 2022) [38] and KEGG database (https://www.kegg.jp/) (accessed on 20 March 2022) were used to annotate the GO and KEGG functions of *LiNAC* gene. The results were collated using the eggNOG-mapper Helper function of TBtools v1.098661 and the text files for downstream analysis were output into GO Enrichment and KEGG Enrichment Analysis for enrichment analysis, respectively. Finally, the data were seen and examined using the online charting tool HIPLOT (https://hiplot.com.cn/) (accessed on 20 March 2022).

### 4.5. Determination of Lignin Content

The plant materials were collected from the greenhouse of Zhejiang Agriculture and Forestry University, which is situated at 30°15′2′′ N/119°43′37′′ E, by upright *L. indica* ‘Shaoguifei’ (Figure 6A) and weeping *L. indica* ‘Xiariwuniang’ (Figure 6B). They were relocated into a habitat with stable temperature and favorable growing circumstances. By keeping track of processes since the time of budding, the growth condition of branches was studied.

The whole branches of ‘Shaoguifei’ and ‘Xiariwuniang’ were weighed 0.5 g at the early growth stage, growth stage, and the end of the growth stage, respectively. Further, three samples of each were collected and ground in liquid nitrogen and the ground powder was packed into 5 mL of an eluent. We then, centrifuged at 12,000 rpm for 20 min after 30 min of shaking the sample at 28 °C, then discarded the supernatant. After adding 100% methanol, the mixture was shaken for 30 min before being centrifuged at 12,000 rpm for 20 min, with the supernatant being discarded. There were four iterations of this stage. After that, they were dried in an oven set to 80 °C for an overnight period.

Accurately weigh 10 mg (can be recorded repeatedly) of the powder (washed and dried) into a 10 mL tube (the total weight of the powder after drying should also be recorded): first add 1 mL of 2 M HCL, then add 0.1 mL of thioglycolic acid, mix it upside down and evenly, then place it in a boiling water bath and heat it for 8 h. Then it was cooled on ice and centrifuged at 12,800 rpm/4 °C for 20 min and the supernatant was discarded. The precipitate was washed twice with distilled water and the precipitate was dried overnight after centrifugation. Then the precipitate was resuspended in 2 mL of 1 M NaOH, mixed evenly, and slightly shaken at 28 °C to react for 18 h. The precipitate was centrifuged at 12,800 rpm for 20 min. Next, 0.5 mL of the supernatant was put into a new glass test tube, 100 μL of concentrated hydrochloric acid was added to each tube, and the solution was placed in a refrigerator at 4 °C for 4 h (this operation was to precipitate the thioglycolate-bound lignin). The solution was centrifuged at 12,800 rpm/4 °C for 20 min to precipitate 1 mL of 1 M NaOH.

After dilution, a UV spectrophotometer was used to determine the absorbance at A280 nm. Each sample underwent three biological replicates, with NaOH solution serving as the blank control. This lignin determination method is referenced by Xu et al. [21].

### 4.6. Plant Materials, RNA Extraction, and qRT-PCR Analysis

Take the entire branches of ‘Xiariwuniang’ and ‘Shaoguifei’ in the early growth stage, growth stage, and the end of the growth stage to examine the expression patterns of *LiNAC* genes. Three samples of each were collected and the sampling time was 10:00. All samples were immediately frozen in liquid nitrogen and stored at −80 °C until needed for RNA isolation.

Total RNA was extracted from plants according to the Instructions of FastPure^®^ Plant Total RNA Isolation Kit (Vazyme, Nanjing, China). Reverse transcription reference HiScript^®^ Ⅲ All-in-one RT SuperMix perfect for qPCR (Vazyme, Nanjing, China) was used. Then, quantitative real-time PCR (qRT-PCR) analysis was performed on an ABI 7300 real-time PCR instrument (Applied Biosystems, Foster City, CA, USA). The reaction system is as follows: SYBR^®^ Premix Ex TaqTM (TaKaRa, Dalian, China) 5 μL, cDNA 2 μL, forward and reverse primers 0.4 μL each, ddH_2_O 2.2 μL. qRT-PCR reaction procedure is: pre-denaturation at 95 °C for 30 s, denaturation at 95 °C for 5 s, denaturation at 60 °C for 30 s, 40 cycles. Three biological replicates were performed for each sample. The relative expression levels of *LiNACs* genes were analyzed by the 2^−ΔΔCt^ method [39] and the experimental data were analyzed by Excel 2010 and SPSS Statistics 20.0 software (IBM Corporation, Armonk, NY, USA). Finally, 21 pairs of gene-specific primers are shown in Appendix A and *LiEF-1α* [40] was used as the internal reference gene.

## 5. Conclusions

In this study, we identified a total of 21 *LiNAC* genes from the transcriptome data of *L. indica*. We analyzed the physicochemical properties of the 21 LiNAC proteins and all *LiNAC* genes were localized in the nucleus, and *LiNAC7* and *LiNAC16* also had a transmembrane structure at the C-terminus. Amino acid sequence alignment showed that almost all *LiNAC* genes contain a conserved NAM structure, consisting of five substructures, A, B, C, D, and E. The phylogenetic tree constructed with *A. thaliana* classified LiNAC proteins into 2 groups and 14 subgroups. GO and KEGG enrichment analysis also indicated that *LiNAC* genes are involved in plant growth development and metabolic pathways. The lignin content in the branches of upright *L. indica* ‘Shaoguifei’ and weeping *L. indica* ‘Xiariwuniang’ were also measured and it was found that the lignin content in the two branches differed at different growth and development stages. The qRT-PCR analysis showed that the expression of *LiNAC13* and *LiNAC8* in ‘Shaoguifei’ was always higher than that in ‘Xiariwuniang’ during S1–S3, while the relative expression of *LiNAC2* in ‘Shaoguifei’ was lower than that in ‘Xiariwuniang’ in S1–S3, which indicated that *L**iNAC2*, *LiNAC13*, and *LiNAC8* might regulate the weeping traits of *L. indica* through their respective regulatory pathways. However, the specific molecular mechanism of regulation and the downstream genes that jointly regulate the weeping traits need to be further verified. In conclusion, the current study improved our understanding of the role of the *NAC* gene family in weeping traits in *L. indica*.

## Figures and Tables

**Figure 1 plants-11-02168-f001:**
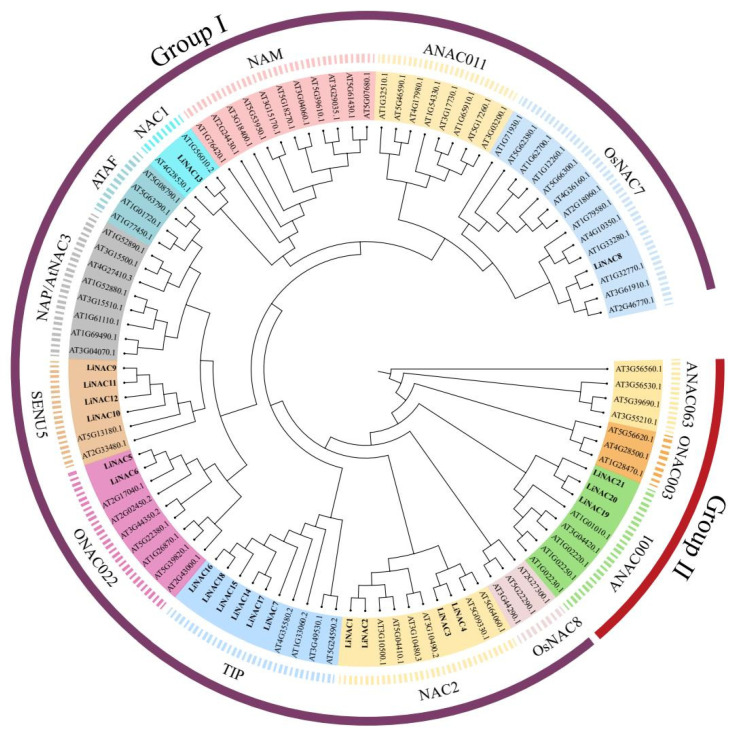
NAC phylogenetic tree of *L. indica* and *A. thaliana*. Each subgroup is distinguished by a different color.

**Figure 2 plants-11-02168-f002:**
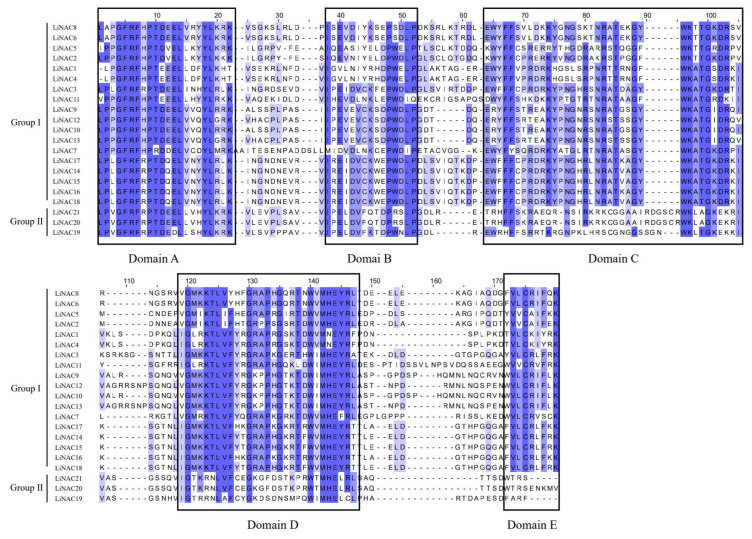
Amino acid sequence alignment of *L. indica*. The black box lines represent the five subdomains (**A**–**E**) of LiNAC.

**Figure 3 plants-11-02168-f003:**
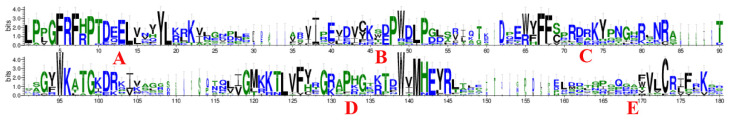
Conserved domain of LiNAC family by Jalview software and Weblogo. (A–E) represent five subdomains.

**Figure 4 plants-11-02168-f004:**
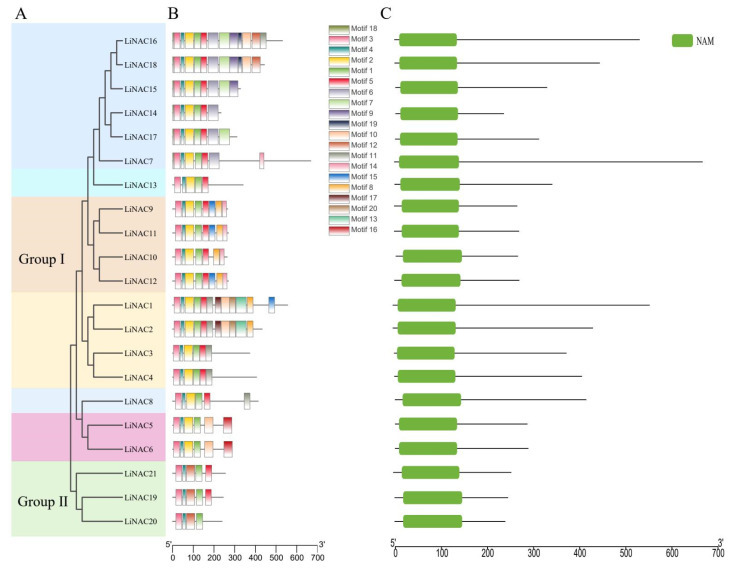
Evolutionary trees (**A**), conserved motifs (**B**), and conserved domain (**C**) of *LiNAC* genes.

**Figure 5 plants-11-02168-f005:**
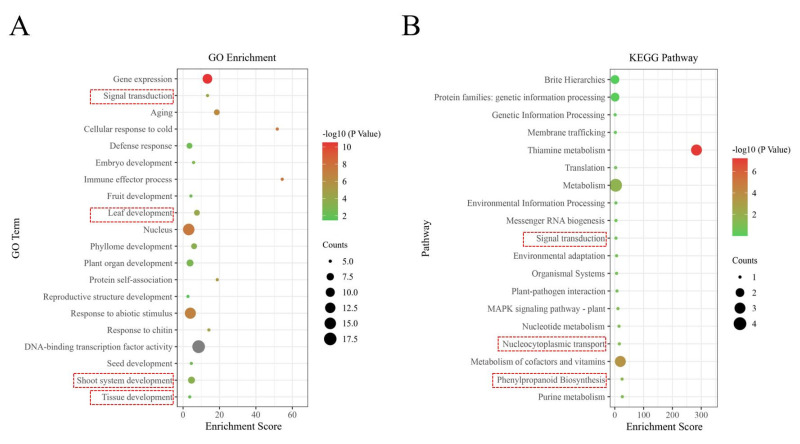
The GO terms (**A**) and KEGG pathways (**B**) enriched of *LiNAC* genes. The black circles indicate the number of target genes and different colors indicate the *p*−value.

**Figure 6 plants-11-02168-f006:**
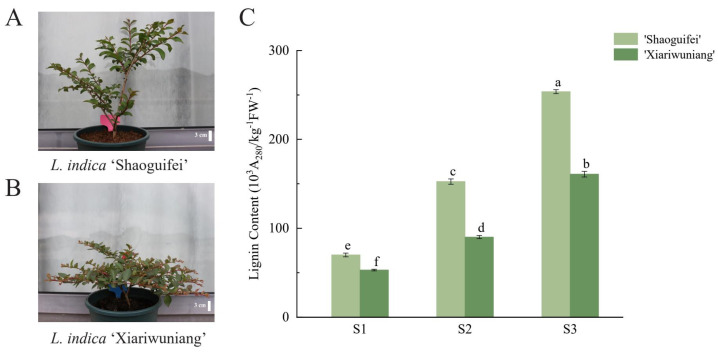
The trait of *L. indica* ‘Shaoguifei’ (**A**) and *L. indica* ‘Xiariwuniang’ (**B**) and lignin content of the two (**C**) at different growth stages. S1, S2, and S3 indicate the early growth stage, the growth stage, and the end of the growth stage. Different letters a–f indicate statistically significant differences in expression.

**Figure 7 plants-11-02168-f007:**
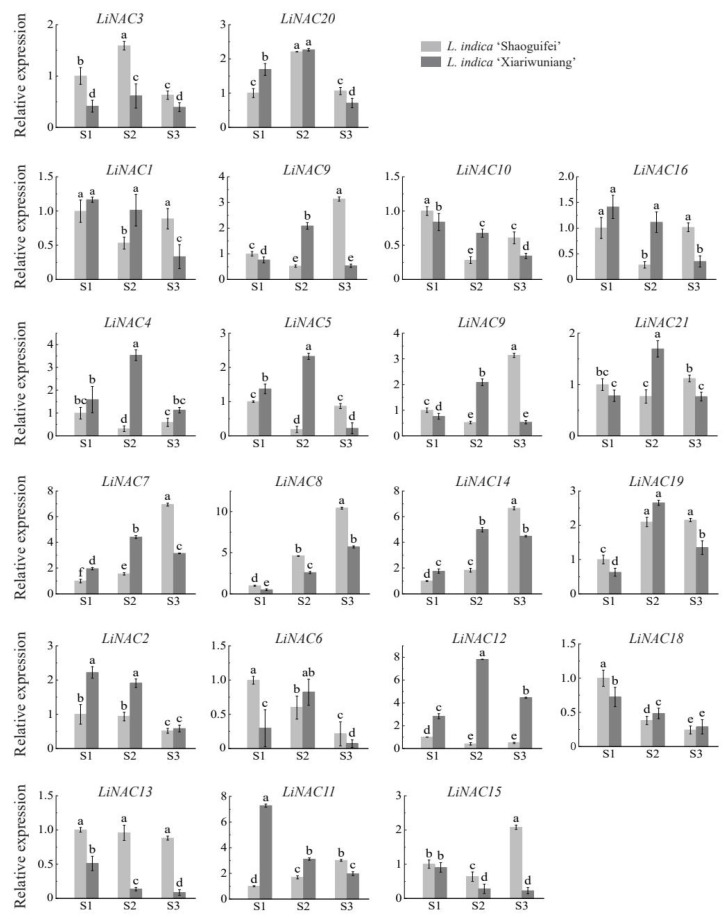
The qRT-PCR analysis of 21 *LiNAC* genes in Upright and Weeping *L. indica* at different growth stages. S1, S2, and S3 indicate the early growth stage, the growth stage, and the end of the growth stage; significant differences are identified by SPSS with Duncan’s test (*p* < 0.05) and are represented by different letters above the error bars.

**Table 1 plants-11-02168-t001:** *LiNAC* gene and protein characterization.

Transcriptome Gene ID (Gene Name)	Amino Acid Number/aa	Molecular Weight/Da	Isoelectric Point	Percentage of the Amino Acids with Highest Content%	Aliphatic Index	Instability Index	GRAVY	Subcellular Localization
TRINITY_DN1513_c0_g1_i2 (*LiNAC1*)	566	61,920.54	4.6	Ser(S)12.2%	65.81	46.22	−0.592	Nucleus
TRINITY_DN1513_c0_g1_i4 (*LiNAC2*)	433	48,711.53	4.49	Ser(S)10.2%	62.84	43.93	−0.758	Nucleus
TRINITY_DN1552_c0_g1_i1 (*LiNAC3*)	373	41,299	4.58	Ala(A)9.7%	69.87	36.19	−0.53	Nucleus
TRINITY_DN1552_c0_g2_i1 (*LiNAC4*)	406	44,609.42	4.42	Gly(G)9.9%	66.6	39.37	−0.525	Nucleus
TRINITY_DN17248_c0_g1_i1 (*LiNAC5*)	286	32,581.02	8.41	Leu(L)10.5%	73.01	48.22	−0.634	Nucleus
TRINITY_DN17248_c0_g1_i5 (*LiNAC6*)	288	32,682.93	8.67	Leu(L)10.4%	69.79	42.84	−0.619	Nucleus
TRINITY_DN1795_c0_g1_i3 (*LiNAC7*)	668	74,336.43	5.15	Ser(S)10.5%	68.64	51.19	−0.691	Nucleus
TRINITY_DN2758_c0_g1_i5 (*LiNAC8*)	414	47,164.68	5.39	Leu(L)8.7%	68.79	59.01	−0.731	Nucleus
TRINITY_DN419_c0_g1_i2 (*LiNAC9*)	266	30,251.89	9.36	Ser(S)10.9%	57.89	52.67	−0.85	Nucleus
TRINITY_DN419_c0_g1_i4 (*LiNAC10*)	264	29,841.31	8.9	Ser(S)12.1%	59.39	56.49	−0.795	Nucleus
TRINITY_DN419_c0_g1_i6 (*LiNAC11*)	270	30,776.37	8.89	Asn(N)& Ser (S) 9.3%	60.63	47.02	−0.905	Nucleus
TRINITY_DN419_c0_g1_i8 (*LiNAC12*)	270	30,781.24	8.89	Ser(S)10.0%	59.92	49.93	−0.942	Nucleus
TRINITY_DN4293_c0_g3_i1 (*LiNAC13*)	341	37,460.37	6.89	Ser(S)10.9%	68.94	49.69	−0.415	Nucleus
TRINITY_DN894_c0_g1_i10 (*LiNAC14*)	234	26,491.05	7.73	Leu(L)11.1%	75.47	29.93	−0.671	Nucleus
TRINITY_DN894_c0_g1_i11 (*LiNAC15*)	328	37,132.65	4.83	Asp(D)8.5%	68.69	38.22	−0.666	Nucleus
TRINITY_DN894_c0_g1_i22 (*LiNAC16*)	531	59,644.15	5.3	Ser(S)8.7%	71.62	48.59	−0.63	Nucleus
TRINITY_DN894_c0_g1_i31 (*LiNAC17*)	311	34,893.12	4.92	Pro(P)8.4%	67.46	47.53	−0.7	Nucleus
TRINITY_DN894_c0_g1_i4 (*LiNAC18*)	444	49,822.84	5.02	Pro(P)8.1%	69.19	47.22	−0.703	Nucleus
TRINITY_DN909_c0_g1_i1 (*LiNAC19*)	245	27,455.47	9.93	Arg(R)10.2%	71.18	62.67	−0.638	Nucleus
TRINITY_DN909_c0_g1_i3 (*LiNAC20*)	239	26,992.33	9.16	Ser(S)11.7%	63.18	58.10	−0.743	Nucleus
TRINITY_DN909_c0_g2_i1 (*LiNAC21*)	255	28,368.91	9.3	Ser(S)11.4%	59.65	55.76	−0.747	Nucleus

## Data Availability

All data in this study can be found in the manuscript or in the Appendix A.

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
