# Peer review of "Identification and Expression Analysis of NAC Gene Family in Weeping Trait of Lagerstroemia indica"

_plants, 2022, doi:10.3390/plants11162168_

Round 1

Reviewer 1 Report

The Authors have described in detail transcripts of 21 NAC family genes from two cultivars of Lagerstroemia indica, however it is not clear whether sequencing data they used comes from these cultivars or not. Only the database in general is indicated as a source of transcriptome sequencing data.    If sequences do not derive from these two cultivars, the differences in qPCR results can come from mismatches between mRNA templates and qPCR primers, and conclusions can be misleading.  

It would be much better to compare whole transcriptomes from the two cultivars to look at differences in expression of other genes involved in cell wall synthesis. Still this will be only the correlation of expression of some genes with weeping and nothing more.

In methods section – lignin content assay should be described in detail, since the paper cited (20) cannot be found.

In discussion Authors say that: ”In this study, qRT-PCR analysis of upright L. indica ‘Shaoguifei’ and weeping L. indica ‘Xiariwuniang’ at three different growth stages showed that LiNACs played an important role in weeping trait of L. indica” while Authors have only compared expression of 21 genes, they have chosen as  likely “important”. Moreover it is not clear whether these transcripts can be found in the two cultivars. Are the 21 NAC transcripts in the two cultivars identical? Are differences in expression of homologous genes connected somehow with the structure of their genes or not, or this is not known yet?

Reviewer 2 Report

The manuscript titled “Identification and expression analysis of NAC gene family in Weeping trait of Lagerstroemia indica” provides novel findings. However, two major concerns which needs to be addressed are:

1.     Pleaser revise the manuscript for large number of typos

2.     Limited literature review search was done, many new studies were not cited or used for this study, please go through them, and cite appropriately.
